# Development and Validation of the Serious Educational Game in Nursing Appraisal Scale

**Carla Sílvia Fernandes** [1,*] 📷, **Maria Joana Campos** [1] 📷, **Maria Teresa Moreira** [2] 📷, **Andreia Lima** [3] 📷, **Salomé Ferreira** [4] 📷 **and Maria Manuela Martins** [5] 📷

[1] CINTESIS@RISE and Nursing School of Porto (ESEP), 4200-072 Porto, Portugal; joana@esenf.pt
[2] CINTESIS@RISE and Health Sciences School-Fernando Pessoa, 4200-256 Porto, Portugal; tmoreira@ufp.edu.pt
[3] CINTESIS@RISE and Higher School of Health at Polytechnic Institute of Viana do Castelo,
4900-347 Viana do Castelo, Portugal; amarialima@ess.ipvc.pt
[4] Higher School of Health at Polytechnic Institute of Viana do Castelo, 4900-347 Viana do Castelo, Portugal;
salomeferreira@ess.ipvc.pt
[5] ICBAS-Abel Salazar Institute of Biomedical Sciences, 4050-313 Porto, Portugal; mmmartins@icbas.up.pt
[*] Correspondence: carlafernandes@esenf.pt

**Abstract:** Objectives: This study aims to develop and validate the Serious Educational Game in Nursing Appraisal Scale (SEGiNAS), a tool designed to evaluate the implementation of serious games within nurse education contexts of quantity of process, quality of process, and learning outcomes. Methods and Materials: This methodological and psychometric study aimed to develop and validate a scale. The item generation phase was based on the cognitive theory of multimedia learning, resulting in a 20-item scale. The validation phase involved evaluating the psychometric scale by surveying 160 Portuguese nurses. Results: A factor analysis revealed a three-factor structure corresponding to the scale's designed dimensions, explaining a total variance of 64.5%. The scale demonstrated high internal consistency for all factors, including engagement and teaching effectiveness (0.925), learning impact and practical application (0.883), and content relevance and clarity (0.848). The dimensions were engagement and teaching effectiveness, learning impact and practical application, and content relevance and clarity. Conclusions: The SEGiNAS scale represents a valid and reliable tool for evaluating serious games in nursing education. Its development fills an existing gap in assessing the teaching–learning process with serious games. This study was not registered.

**Keywords:** serious game; nurse; psychometrics; scale; validation study

## 1. Introduction

Gamification represents a particularly new and appealing strategy [1], offering a motivational approach to engage the player [2]. These strategies utilize points, levels, badges, leaderboards, and challenges to hold students' attention and promote a more engaging educational environment [1–3]. Evidence suggests that gamification offers specific advantages, such as allowing students to progress at their own pace, receive immediate feedback, engage in a competitive dimension, stay motivated, and engage in a free environment. No evidence suggests that gamification impairs the learning process or participant satisfaction [3]. Unlike traditional methods, which may fail to engage students fully, gamification introduces fun, competition, and reward elements, thus increasing student engagement and retention [4]. With new generations of students entering the educational system and demanding new learning strategies like gamification, the importance of traditional didactics seems to be diminishing [1].

In recent years, gamification has aroused considerable interest in various fields, including health [5–8]. Moreover, the diffusion of technological and computer tools has contributed to the use of new methodologies, strategies, and resources in teaching [2,5], specifically in the education of healthcare professionals [2]. In the education of healthcare

professionals, gamification emerges as an innovative strategy that has sparked the interest of teachers [1–3]. However, many teachers feel challenged in the practical implementation of this strategy and in evaluating its effectiveness and impact on the teaching–learning process [5].

The strategic use of gamification in healthcare professions education aims to improve learning outcomes, promote behavioral changes, and enhance clinical skills through greater engagement and practice in a risk-free environment [6,7]. Within this context, exploring gamification in the teaching–learning process opens various innovative possibilities, among which we highlight the use of serious games. Considering the previously discussed themes, the recourse to serious games in education aims to motivate students in the teaching–learning process, demanding from its developers an additional study on gamification and pedagogy. This effort is necessary to transform the process into interactive and captivating experiences, contributing to its effectiveness and preparing students for real-life challenges [9].

A "serious game" is created with a primary goal beyond entertainment, focusing on specific objectives such as education, training, or health promotion [9,10]. Using these resources, the goal is to engage players in a manner that is both enjoyable and effective to achieve concrete results in terms of learning or behavioral change [11]. Serious games can take many forms, such as simulations, role-playing games (RPG), educational games, and physical games like cards and board games. They can be used in various environments, including classrooms, workplaces, and health contexts [11,12].

Serious games are increasingly being adopted in secondary education for nursing practice, presenting valuable learning strategies that engage students and staff. These games effectively differentiate between theoretical knowledge and practical application, smoothly integrating the two. This approach enhances learning engagement and bridges the gap between classroom theories and real-world nursing skills. Adapting to evolving healthcare needs requires access to qualified and adaptable nurses [13]. It is essential to develop robust assessment tools to ensure that serious games meet their educational objectives. These tools are crucial for accurately measuring the impact of games on learning processes and outcomes across various types [14,15]. The direct influence of educational quality on patient care quality underscores the importance of validating these tools. This validation is vital to assess how innovative learning styles, such as serious games, significantly enhance the skill development of health professionals, particularly in nursing [13,16,17].

Integrating Richard Mayer's cognitive theory of multimedia learning into teaching and learning process assessment, especially in health education, emphasizes the importance of dual-channel, limited capacity, and active processing in effective educational strategies, including serious games. He outlines the development of the theory, focusing on dual-channel processing (separate channels for verbal and visual information), limited capacity (processing capacity is severely limited), and active processing (meaningful learning involves selecting relevant material to be processed in working memory, organizing it into coherent verbal and visual structures, and integrating it with relevant knowledge from long-term memory) (Figure 1) [18–20]. This framework is crucial for assessing the teaching–learning process, focusing on the quality and quantity of the process and understanding learning outcomes in terms of acquisitions, effects, and visual representations. This theory's understanding [18] of optimizing verbal and visual information processing can guide the development and validation of instruments to measure the quality of processes and outcomes of serious games, connecting the existing gap at this level. Aware of this need, this study aims to develop and validate the Serious Educational Game in Nursing Appraisal Scale (SEGiNAS), a tool designed to assess the implementation of serious games in nursing education contexts in the quantity of process, quality of process, and learning outcomes.

**Figure 1.** Richard Mayer's Cognitive Theory of Multimedia Learning used in Scale Development.

## 2. Materials and Methods

### 2.1. Study Design

We conducted a cross-sectional psychometric evaluation study aimed at developing and validating the Serious Educational Game in Nursing Appraisal Scale (SEGiNAS), a tool for assessing the use of serious games in health education, including the quantity of process, quality of process, and learning outcomes.

### 2.2. Item Generation

The cognitive theory of multimedia learning [18] supported the scale's construction. The 20 scale items were methodically segmented into categories reflecting key aspects of the learning process, ensuring a comprehensive assessment tool focused on three main elements, quantity of process, quality of process, and learning outcomes.

Firstly, the quantity of process includes four items to ensure that the strategy used matches the intended objectives and the duration of the pedagogical activity. The quality of process evaluates the content design through four items focused on structure, clarity, motivation, participation, and methodology. Lastly, the scale integrates items for the assessment of learning outcomes in three dimensions with 12 items, including acquisition outcomes (what students gained), effects (application of skills in the workplace), and outcome images (students' perceptions and attitudes toward the training). The 20 items of the proposed scale were developed based on a detailed review of the cognitive theory of multimedia learning. This initial phase allowed for the construction of the Serious Educational Game in Nursing Appraisal Scale (SEGiNAS), with 20 items for the validation phase and a Likert scale from 1 to 5 (1: strongly disagree, 2: disagree, 3: neutral, 4: agree, 5: strongly agree), for each item.

### 2.3. Participants and Data Collection

The sample consisted of a total of 160 Portuguese nurses. The invitation to participate in this study was made through its announcement on the website of the Portuguese Nursing Association (Ordem dos Enfermeiros), where nurses expressed their response to the invitation. The inclusion criteria were limited to nurses actively practicing in Portugal and registered with the Nursing Association; this included only those proficient in Portuguese. Excluded were nursing students, Portuguese nurses practicing abroad, nurses in Portugal without language proficiency, and those who declined to sign the informed consent. A non-probabilistic convenience sampling was used, given that this study's announcement was made on their website, using social networks as data collection sources. Portuguese nurses were selected as part of the inclusion criteria. Participants were required to download a serious game [7] from the Google Play Store and assess it by completing a questionnaire on Google Forms®. The evaluation was conducted using the newly developed SEGiNAS scale. In addition to the scale, sociodemographic and professional data of the participants, such as gender, age, academic qualifications, and professional category, were requested.

### 2.4. Data Analysis

The Statistical Package for the Social Sciences (SPSS), version 29, was used for data processing. Regarding psychometric properties, the guidelines suggested in the specific literature were adopted, focusing on evaluating the reliability and validity of measurement instruments [21–23]. The Kaiser–Meyer–Olkin index was used to assess the adequacy of the sample for factorization and the performance of a principal component analysis with

orthogonal varimax rotation. The factor analysis was conducted following Kaiser's rule, excluding factor loadings below 0.40 when present. Furthermore, a total explained variance above 40% was sought, and the internal consistency was checked through Cronbach's alpha coefficient.

### 2.5. Ethical Considerations

Approval was obtained before the start of this study. The research adhered to ethical principles and was approved by the Research Ethics Committee in April 2021 (ADESEP 622021). A virtual informed consent form was incorporated into the first section of the questionnaire. This involved a page describing the research and requesting permission to use the data.

## 3. Results

### 3.1. Scale

During the validation process of a 20-item scale, 160 nursing professionals participated. Of this group, 86.9% were women, with an average age of 38.6 years (standard deviation of $\pm 9.3$), ranging from 23 to 61 years. Almost all nurses ($n = 155$ or 96.9%) hold bachelor's degrees, with only one nurse having an associate degree, four with master's degrees, and none with doctorates. The majority of the nurses ($n = 110$ or 69%) are classified as staff nurses, followed by specialist nurses ($n = 44$ or 28%) and finally, head nurses ($n = 6$ or 4%). The average professional experience is 16.2 years (standard deviation of $\pm 9.6$), ranging from 1 to 37 years.

The stepwise results of the psychometric evaluation will be proposed in the following. Table 1 shows the details of the participants' responses. The options on the Likert scale were identified by their numerical positions, ranging from 1 to 5, indicating that the answers cover the entire possible spectrum of the scale.

**Table 1.** Descriptive statistical analysis of the scale.

| Item | Strongly Disagree | Disagree | Neutral | Agree | Strongly Agree |
|---|---|---|---|---|---|
| Using this strategy contributes to a better understanding of the contents | 0.0 | 0.6 | 8.8 | 56.0 | 34.6 |
| The content covered in this serious game was useful for developing my professional activity | 0.0 | 2.5 | 10.7 | 56.0 | 30.8 |
| The time allocated to the applicability of this serious game is adequate | 0.6 | 6.9 | 6.3 | 59.1 | 27.0 |
| The themes of this serious game are presented clearly and coherently | 0.0 | 1.9 | 6.3 | 52.2 | 39.6 |
| The content of this serious game is appropriate between theory and practice | 0.0 | 1.3 | 6.3 | 57.9 | 34.6 |
| Learning from this serious game will impact my performance in practice | 0.0 | 1.9 | 13.2 | 57.2 | 27.7 |
| Using this serious game contributes to the motivation of the trainees | 0.0 | 0.6 | 9.4 | 44.7 | 45.3 |
| I feel an evolution of my knowledge in the area with the use of this serious game | 0.0 | 1.9 | 13.8 | 56.6 | 27.7 |
| I intend to apply the knowledge acquired with this serious game | 0.0 | 0.0 | 10.7 | 56.0 | 33.3 |
| The content developed in this serious game was in an adequate amount for my level of knowledge | 0.6 | 3.1 | 6.3 | 61.0 | 28.9 |
| The moment provided by this serious game encouraged the participation of the trainees | 0.0 | 0.6 | 3.8 | 45.3 | 50.3 |
| I think this strategy is useful for learning | 0.0 | 0.0 | 5.7 | 43.4 | 50.9 |

**Table 1.** *Cont.*

| Item | Strongly Disagree | Disagree | Neutral | Agree | Strongly Agree |
|------|------|------|------|------|------|
| The duration of this serious game is adequate to allow learning | 0.0 | 3.8 | 11.9 | 50.9 | 33.3 |
| I would recommend the content covered in this serious game to others | 0.0 | 0.6 | 6.3 | 44.7 | 48.4 |
| This serious game is useful for reviewing knowledge | 0.0 | 0.0 | 6.9 | 42.1 | 50.9 |
| Using this serious game is a good resource to spark interest in the topic | 0.0 | 0.0 | 8.2 | 45.9 | 45.9 |
| This serious game is useful for diagnosing gaps | 0.0 | 0.6 | 9.4 | 52.8 | 37.1 |
| This serious game helps to retain specific knowledge | 0.0 | 0.0 | 8.2 | 54.7 | 37.1 |
| This serious game is an appropriate teaching strategy for skill acquisition | 0.0 | 1.3 | 10.1 | 54.7 | 34.0 |
| Overall, the use of this serious game was satisfactory for me | 0.0 | 0.0 | 5.7 | 48.4 | 45.9 |

### 3.2. Factor Analysis

After individually studying the scale's items, a conceptual structure analysis of this scale was conducted through a factor analysis to identify the underlying factors in participants' responses. Firstly, to check if these data are appropriate for conducting a factor analysis, the correlation matrix among responses to questions was analyzed, observing moderate and high correlations. Furthermore, the Kaiser–Meyer–Olkin measure of sampling adequacy was calculated, resulting in a value of 0.944.

The results of the forced three-factor analysis (explaining 64.5% of the total variance), followed by varimax rotation and Kaiser normalization, are presented in Table 2, which indicates the factor weights of different items on each factor, with the highest weight of each item highlighted in bold.

**Table 2.** Factorial structure of the scale and communalities of factors.

| Item | Factor 1 | Factor 2 | Factor 3 | Communalities |
|------|------|------|------|------|
| 1 | 0.176 | 0.349 | **0.654** | 0.581 |
| 2 | 0.226 | 0.309 | **0.697** | 0.633 |
| 3 | 0.212 | −0.018 | **0.786** | 0.664 |
| 4 | 0.332 | 0.396 | **0.573** | 0.596 |
| 5 | 0.311 | 0.442 | **0.601** | 0.654 |
| 6 | 0.190 | **0.647** | 0.455 | 0.663 |
| 7 | **0.575** | 0.302 | 0.460 | 0.633 |
| 8 | 0.274 | **0.711** | 0.337 | 0.694 |
| 9 | 0.384 | **0.671** | 0.354 | 0.723 |
| 10 | 0.282 | **0.704** | 0.130 | 0.592 |
| 11 | **0.560** | 0.465 | 0.218 | 0.578 |
| 12 | **0.605** | 0.460 | 0.300 | 0.668 |
| 13 | 0.355 | 0.371 | **0.455** | 0.471 |
| 14 | **0.776** | 0.225 | 0.188 | 0.687 |

**Table 2.** *Cont.*

| Item | Factor 1 | Factor 2 | Factor 3 | Communalities |
|------|----------|----------|----------|---------------|
| 15 | **0.755** | 0.256 | 0.292 | 0.721 |
| 16 | **0.806** | 0.208 | 0.278 | 0.770 |
| 17 | 0.434 | **0.482** | 0.263 | 0.490 |
| 18 | 0.567 | **0.603** | 0.151 | 0.709 |
| 19 | **0.569** | 0.568 | 0.183 | 0.680 |
| 20 | **0.729** | 0.317 | 0.264 | 0.702 |

Table 2 presents the commonalities, including the percentage of each variable's (item's) variance explained jointly using the three extracted factors. This percentage is close to 50% for almost all items. Items 13, 17, 18, and 19 were maintained and allocated to their respective factors based on the theoretical framework that guided the scale's construction.

For a straightforward interpretation, the items are regrouped in the following table (Table 3) according to the highest factorial loading of each to make it possible to visualize which items saturate each factor and thus identify the underlying dimensions of the responses. Three factors are identified, corresponding to different dimensions of the scale, as follows: "Factor 1—Engagement and Teaching Effectiveness", which includes eight items; "Factor 2—Learning Impact and Practical Application", covering six items; and lastly, "Factor 3—Content Relevance and Clarity", also with six items.

**Table 3.** Scale dimensions.

| Factor 1—Engagement and Teaching Effectiveness<br>Item | Factorial Loading |
|---|---|
| The use of this serious game contributes to the motivation of the trainees | 0.575 |
| The opportunity provided by this serious game encouraged the participation of the trainees | 0.560 |
| Considers this strategy useful for learning | 0.605 |
| Would recommend the content covered in this serious game to others | 0.776 |
| This serious game is useful for reviewing knowledge | 0.755 |
| The use of this serious game is a good resource to spark interest in the topic | 0.806 |
| This serious game is an appropriate teaching strategy for skills acquisition | 0.569 |
| Overall, the use of this serious game was satisfactory to me | 0.729 |
| **Factor 2—Learning Impact and Practical Application**<br>**Item** | **Factorial Loading** |
| Believes that learning from this serious game will impact their performance in practice | 0.647 |
| Felt an evolution in their knowledge in the field with the use of this serious game | 0.711 |
| Intends to apply the knowledge acquired with this serious game | 0.671 |
| The content developed in this serious game was in an adequate quantity for their level of knowledge | 0.704 |
| This serious game is useful for diagnosing gaps | 0.482 |
| This serious game helps to retain specific knowledge | 0.603 |
| **Factor 3—Content Relevance and Clarity**<br>**Item** | **Factorial Loading** |
| The use of this serious game is a good resource to spark interest in the topic | 0.654 |
| Considers the content covered in this serious game useful for the development of their professional activity | 0.697 |
| The time allocated for the applicability of this serious game is appropriate | 0.786 |
| The themes of this serious game are presented in a clear and coherent manner | 0.573 |
| The content of this serious game is suitable between theory and practice | 0.601 |
| The duration of this serious game is appropriate to allow for learning | 0.455 |

*3.3. Internal Consistency*

Subsequently, the questionnaire's internal consistency was checked using Cronbach's alpha coefficients. Table 4 presents the values of this coefficient for the identified dimensions, concluding that the consistency of all dimensions is good.

**Table 4.** Cronbach's alpha of the scale dimensions.

| Dimension | Alpha |
| --- | --- |
| 1—Engagement and Teaching Effectiveness | 0.925 |
| 2—Learning Impact and Practical Application | 0.883 |
| 3—Content Relevance and Clarity | 0.848 |

## 4. Discussion

This study aimed to develop and validate the Serious Educational Game in Nursing Appraisal Scale (SEGiNAS), a tool designed to evaluate the implementation of serious games within nurse education, focusing on aspects such as quantity of process, quality of process, and learning outcomes. SEGiNAS emerges as a pioneering tool, filling gaps by assessing serious games in the context of nursing education. The development and validation of the Serious Educational Game in Nursing Appraisal Scale (SEGiNAS) marks a significant improvement at the intersection of serious game use and nursing education. This initiative responds to the growing attention given to serious games in nursing [3,13,16], supported by changing learning preferences and the search for innovative teaching methodologies. Technological progress highlights the shortcomings of traditional education, stimulating reforms toward involving methods like gamification in nursing education to better meet training needs [24]. Traditional educational methods, such as PowerPoint® presentations, lectures, and online didactic modules that are commonly used to deliver content to professional nurses, often provide limited student engagement and interaction, potentially compromising content retention [3,25]. Despite growing interest in gamification, specifically in serious games in health, a significant gap persists in evaluating and validating these strategies, underlining a crucial need for more rigorous studies to determine their effectiveness and medium- and long-term implementation strategies [11]. The available evidence on the use of serious games indicates positive outcomes, necessitating the development of precise measures to effectively assess increases in student satisfaction, engagement, and knowledge retention [1,3,10,11,16]. The field of serious games in health is marked by a notable absence of rigorous evaluation tools, highlighting an urgent need for comprehensive methodologies to comprehensively validate their impact and efficacy [11].

The cognitive theory of multimedia learning guided the development of the SEGiNAS tool; this theoretical framework emphasizes optimizing the processing of verbal and visual information, which is essential for assessing the quality and outcomes of the teaching–learning process [18].

The development and validation steps of SEGiNAS involved a comprehensive approach to item generation, reflecting the main aspects of the learning process regarding the quantity and quality of the teaching–learning process and learning outcomes. The validation process demonstrated the scale's robust psychometric properties, including high internal consistency in its three dimensions, engagement and teaching effectiveness, learning impact and practical application, and content relevance and clarity.

In the dimension of engagement and teaching effectiveness, the factor underscores serious games' motivational and engaging aspects, emphasizing the importance of capturing students' interest and promoting active participation. The challenge is to develop strategies that increase motivation, engagement, and attention, which are substantial elements for learning improvement [26]. Items in this dimension also include evaluating the strategy's effect on knowledge and learning. Assessing teaching effectiveness and objectively measuring learning outcomes are crucial for improving the quality and effectiveness of

teaching strategies [27]. This dimension reflects the cognitive theory of learning approach, emphasizing the engagement and interaction required for effective learning [18].

In the dimension learning impact and practical application, which includes six scale items, this factor focuses on the outcomes of using serious games, including knowledge acquisition, skills application in practice, and content retention. In nursing education, developing critical thinking, clinical skills, clinical knowledge, communication, ethics, accountability, and lifelong learning as essential competencies for nursing graduates is imperative, underlining the need for suitable evaluation tools to facilitate more effective teaching [27]. Assessing the practical application of skills and knowledge reconciles the principles of the cognitive theory of learning [19,20] with the meaningful organization and integration of information into students' cognitive structures, ensuring that learning is retained and readily applicable.

The last dimension, content relevance and clarity, includes six items related to time, themes, content, duration, and the utility of the serious game. In the design and development of a serious game, it is crucial to organize content progressively and challengingly, ensure alignment with educational objectives, incorporate reflection and feedback within narrative environments, and consider the time, themes, duration, and utility to achieve and engage students in the proposed educational objectives [9].

Finally, both the questionnaire and the identified dimensions reveal good internal consistency and reliability of the scale, meaning they are valid for the intended purposes, allowing confidence in the conclusions and results.

### 4.1. Limitations

This study has some limitations, one related to the sampling method. The use of convenience sampling, although practical, may only partially represent the diversity of the nursing professional population. Moreover, this study was conducted in a specific cultural and professional environment (Portuguese nurses), which may influence the applicability of SEGiNAS in different cultural or educational contexts. Furthermore, as previously mentioned, items 13, 17, 18, and 19 were retained and assigned to their respective factors based on the theoretical framework, which could be considered a limitation.

### 4.2. Implication for Practice

The development and validation of SEGiNAS marks a significant milestone in nursing education, serious game design, and research. SEGiNAS addresses a fundamental requirement by offering a validated tool for assessing and evaluating the design of serious games, thereby enhancing their effectiveness and relevance in educational settings. It may also have significant implications for nursing education and designers and developers of serious games for educational purposes, ensuring that these strategies are designed for a "serious" effect and not merely for "play".

## 5. Conclusions

SEGiNAS provides a robust framework for educators to assess the quality and effectiveness of serious games comprehensively; the scale and the identified dimensions reveal good internal consistency and reliability. This pathway can improve pedagogical strategies, making them more engaging, interactive, and tailored to nursing students' learning outcomes. Moreover, it can guide the design of serious games, ensuring they are effectively aligned with educational objectives and the needs of students.

**Author Contributions:** Conceptualization, C.S.F., M.J.C. and M.M.M.; methodology, C.S.F., M.J.C., M.T.M., A.L., S.F. and M.M.M.; formal analysis, C.S.F. and A.L.; writing—original draft preparation, C.S.F., A.L. and M.T.M.; writing—review and editing, M.J.C., S.F. and M.M.M.; supervision, M.M.M. All authors have read and agreed to the published version of the manuscript.

**Funding:** This research received no external funding.

**Institutional Review Board Statement:** This study was conducted in accordance with the Declaration of Helsinki and approved by the Ethics Committee code ADESEP 622021 of approval for studies involving humans in April 2021.

**Informed Consent Statement:** Informed consent was obtained from all subjects involved in this study.

**Data Availability Statement:** The dataset is available on request from the authors.

**Public Involvement Statement:** No public involvement in any aspect of this research.

**Guidelines and Standards Statement:** Recommendations for reporting the results of studies of instrument and scale development and testing.

**Use of Artificial Intelligence:** AI or AI-assisted tools were not used in drafting any aspects of this manuscript.

**Conflicts of Interest:** The authors declare no conflicts of interest.

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
