# Peer review of "Development and Validation of the Serious Educational Game in Nursing Appraisal Scale"

_nursrep, doi:10.3390/nursrep14020087_

Round 1

Reviewer 1 Report

Comments and Suggestions for Authors

This manuscript presents a design and validation study of a scale to measure the use of serious games in nursing. The topic is very interesting and necessary, as there is a lack of instruments on this subject. Serious games in nursing are a very successful methodology for teaching but, as the authors point out, there are no validated tools to assess whether their use has been successful. This study fills this need.

The resulting instrument is the result of a rigorous application of the scientific method. The proposed objectives were fully met, resulting in an easily applicable and effective product. The authors carry out a thorough analysis of the instrument and provide evidence of its validity and reliability.

I would like the authors to add the instrument with the likert scale designed for its application as an appendix at the end of the manuscript.

Author Response

Dear Editor and Reviewers,

We are grateful for the feedback and suggestions made during our manuscript review.  The changes made have improved the quality of our manuscript, and below we provide our point-by-point response to each of the comments.

Reviewer1:

The resulting instrument is the result of a rigorous application of the scientific method. The proposed objectives were fully met, resulting in an easily applicable and effective product. The authors carry out a thorough analysis of the instrument and provide evidence of its validity and reliability.

I would like the authors to add the instrument with the likert scale designed for its application as an appendix at the end of the manuscript.

Thanks for the comment. Attached Appendix I

Reviewer 2 Report

Comments and Suggestions for Authors

Introduction:

Suggest changing the third sentence to: Evidence suggests gamification offer specific advantages.... and ending the sentence with free environment. Start next sentence with "To date, there is no evidence to suggest gamification in any way impairs the learning process...."

Second paragraph, remove 'in the education of healthcare professionals from the end of the sentence as this is repetitive with the beginning of the next sentence.

Combine the paragraph starting with "Thus, the emergence of effective assessment tools become essential"....with the paragraph above. Also add the next paragraph to either the one above or below as it is only 2 sentences and paragraphs need to be at least 3 sentences.

Suggest adding some kind of graphic to represent the theory. This is helpful to the reader to visualize the concepts and how it relates to your study.

Is it possible to add a bit more detail such as if they played the game on a mobile app (their phone) or computer etc?

Author Response

Dear Editor and Reviewers,

We are grateful for the feedback and suggestions made during our manuscript review.  The changes made have improved the quality of our manuscript, and below we provide our point-by-point response to each of the comments.

Reviewer2:

Suggest changing the third sentence to: Evidence suggests gamification offer specific advantages.... and ending the sentence with free environment. Start next sentence with "To date, there is no evidence to suggest gamification in any way impairs the learning process...." Thanks for the comment. Changed

Second paragraph, remove 'in the education of healthcare professionals from the end of the sentence as this is repetitive with the beginning of the next sentence. Thanks for the comment. Changed

Combine the paragraph starting with "Thus, the emergence of effective assessment tools become essential"....with the paragraph above. Also add the next paragraph to either the one above or below as it is only 2 sentences and paragraphs need to be at least 3 sentences. Thanks for the comment. Changed

Suggest adding some kind of graphic to represent the theory. This is helpful to the reader to visualize the concepts and how it relates to your study. (Figure 1) Added

Reviewer 3 Report

Comments and Suggestions for Authors

Dear authors,

thanks for the opportunity to review your very interesting manuscript. I´d like to share my thoughts with you on the content and provided information and point out some inambiguities to be considered. I hope this will be helpful for guiding your revisions of the manuscript. 

ABSTRACT: The scale is named "SEInGAS", but I guess the appropriate akronym based on the phrases behind this akronym would be "SEGiNAS"?

1 INTRODUCTION SECTION:

Well written, the provided backround is comprehensible and the gap ist clearly stated. The theoretical considerations of the positives of gamification theory and their didactic implications are well described.

Para 1 (p.3): I´m not sure if the aim is to assess "the implementation". In my understanding, the instrument should assess "Quantity of Process, Quality of Process, and Learning Outcomes" of SEGs (as stated in sect. 2.2) and therefor you should change the aim of the study to be more consistent. Overall (and considering the content of the single items) the scale seems to aim to evaluate the appropriateness/suitability of certain SEGs for learning in nursing...this is not really reflected by the provided aims so far...

2 MATERIALS AND METHODS SECTION

2.1: please shorten and change to (e.g.): We conducted a cross-sectional psychometric evaluation study.

2.2: please provide some additional information on the item construction and item selection process; who was involved? how were the items established (literature?)? was there initially an even bigger pool of items? Have there been any additional efforts or tests (e.g. content validation, expert discussions, delphi technique...) to safeguard the relevance and clarity of the established items before testing it on the target population?

2.3: change to "consisted of a total of 160 Portuguese nurses".

2.3: Please provide any additional information related to inclusion/exclusion criteria!

2.3: "...evaluate the game by filling out a questionnaire on Google Forms®" --> I guess they precisely used the newly developed "SEInGAS"? please state clearly!

2.3: Please provide extended information about the downloaded SEG! What was its nursing related content/learning focus/learning outcomes? What was its duration/frequancy of use? What was the design of this SEG?

2.4: "The Kaiser-Meyer-Olkin index was used to assess the adequacy of simple correlations" change to "Kaiser-Meyer-Olkin index was used to assess the adequacy of the sample for factorization"

2.4: Why did you initially use orthogonal rotation? Initially, oblique rotation is suggested for initially factorization...did you try eg PROMAX first? Please state!

2.4: Why did you check Cronbach's alpha for the whole scale? Following the description of the final construct below, the scale seems not to be unidimensional but three-dimensional. So providing a total scale Cronbach's alpha seems not to be appropriate and alphas should be computed for each dimension individually...

3 RESULTS SECTION

3.1, para 2: Change from "The scale's psychometric properties will be analysed in the following paragraphs, specifically focusing on its reliability and validity" to "The stepwise results of the psychometric evaluation will be proposed in the following".

3.1, para 2: You state "including the minimum value (Min), maximum value (Max), median (Med), mode, and coefficient of variation (Coef Var)" --> I can neither find these values in the table nor are they highlighted. Please adapt the table!

3.1, Tab 1: Change header from "Iten" to "Item"

3.1, Tab1: This might result from the translation process from Portuguese to English, but it seems not to be suitable to formulate QUESTIONS, e.g. "Do you think that learning from this serious game will impact your performance in practice" because this can not properly answered by eg "strongly agree" but rather "yes/no"... Reformulate all items that are questions to STATEMENTS and be sure to provide an initial request to the participants what they should do (e.g. "please assess to which degree you agree with the following statements related to the SEG and mark with an "X""). Otherwise, item formulations and respective anwering options are not consistent and not comprehensible!

3.2: I miss a Bartlett test

3.2: delete (redundant!): "which is extremely high. Con-sequently, the factorability of the correlation matrix is excellent, meaning it is very appro-priate to conduct a factor analysis with these data."

3.2, Tab 2: Items 13, 17, 18 and 19 seem to be double-loading items without showing the suggested difference (approx 0,2) between individual factor loadings to clearly distinguish from each other. You should state in the following text section why these items were  not excluded from the scale and why you chose to allocate them to the respective factors finally. Theoretical reasons?

3.2, Tab 3: Change from "Itens" to "Items"

3.2, Tab 3: change from "Factorial Weight" to "Factor loading"

3.1 AND 3.2 (Tables 1 & 3):  Be consistent in the providion of the formulated items, since both tables show the same items but these items are formulated differently!!!

3.3: Change header from "validity" to "Internal consistency"

3.3: Reformulate from "Subsequently, an analysis of the validity of the questionnaire was conducted, i.e., the internal consistency and reliability of the questionnaire used." to "Subsequently, the internal consistency of the questionnaire was checked by using Cronbach's Alpha coefficients."

3.3, Tab 4: Change from "Dimensao" to "Dimension"

3.3, Tab 4: Change from "Alfa" to "Alpha"

3.3 AND Abstract: Use the coefficients provided in Tab 4 in the Abstract and delete the overall Alpha in the entire Manuscript, since this is not appropriate for the scale.

4 DISCUSSION

4, para 1: this whole section is redundant to the introduction and therefor not suitable for the discussion section; Delete it please!

4, para 1: start with "the aim of this study was to..." and then give a brief overview on the results.

4, para 2,3,4,5,6 are well discussed!

4: Consider to discuss eventually the double loadings of the respective items and discuss that these might be a weakness of the scale (?). Also consider to discuss the used rotation method minding my respective critique above.

4.1: Please delete this: "Another area for improvement is related to the methodological focus of the study; the study design is pri-marily focused on validating the assessment tool without delving into the effectiveness of serious games as an educational strategy over time." This is not a weakness or limitation!

4.2: Please reformulate or delete: "The development and validation of SEInGAS address the current lack of validated tools for measuring the effectiveness of innovative teaching methods." because the scale can NOT evaluate the effectiveness of SEGs but evaluate its design and assess the presented learning content with relation to learning objectives...

4.2: delete the last sentence (On the other hand, this study also underscores the importance of using validated assessment instruments to evaluate serious games.), because this is not to be seen in this study!

5 CONCLUSION

Please delete the first two sentences or relocate them to the start of the discussion section, where it might be more suitable.

Start with "SEInGAS provides a robust..."

Comments on the Quality of English Language

Dear authors,

please find comments regarding the quality of English language included within the suggestions section!

Author Response

Dear Editor and Reviewers,

We are grateful for the feedback and suggestions made during our manuscript review.  The changes made have improved the quality of our manuscript, and below we provide our point-by-point response to each of the comments.

Reviewer3:

ABSTRACT: The scale is named "SEInGAS", but I guess the appropriate akronym based on the phrases behind this akronym would be "SEGiNAS"? Thanks for the comment. Changed

1 INTRODUCTION SECTION:

Well written, the provided backround is comprehensible and the gap ist clearly stated. The theoretical considerations of the positives of gamification theory and their didactic implications are well described. Thanks for the comment.

Para 1 (p.3): I´m not sure if the aim is to assess "the implementation". In my understanding, the instrument should assess "Quantity of Process, Quality of Process, and Learning Outcomes" of SEGs (as stated in sect. 2.2) and therefor you should change the aim of the study to be more consistent. Overall (and considering the content of the single items) the scale seems to aim to evaluate the appropriateness/suitability of certain SEGs for learning in nursing...this is not really reflected by the provided aims so far... Thanks for the comment. Changed

2 MATERIALS AND METHODS SECTION

2.1: please shorten and change to (e.g.): We conducted a cross-sectional psychometric evaluation study. Thanks for the comment. Changed

2.2: please provide some additional information on the item construction and item selection process; who was involved? how were the items established (literature?)? was there initially an even bigger pool of items? Have there been any additional efforts or tests (e.g. content validation, expert discussions, delphi technique...) to safeguard the relevance and clarity of the established items before testing it on the target population? Thanks for the comment. Added

2.3: change to "consisted of a total of 160 Portuguese nurses". Thanks for the comment. Changed

2.3: Please provide any additional information related to inclusion/exclusion criteria! Thanks for the comment. Added

2.3: "...evaluate the game by filling out a questionnaire on Google Forms®" --> I guess they precisely used the newly developed "SEInGAS"? please state clearly! Thanks for the comment. Explain

2.4: "The Kaiser-Meyer-Olkin index was used to assess the adequacy of simple correlations" change to "Kaiser-Meyer-Olkin index was used to assess the adequacy of the sample for factorization" Thanks for the comment. Added

2.4: Why did you check Cronbach's alpha for the whole scale? Following the description of the final construct below, the scale seems not to be unidimensional but three-dimensional. So providing a total scale Cronbach's alpha seems not to be appropriate and alphas should be computed for each dimension individually... Thanks for the comment. Changed

3 RESULTS SECTION

3.1, para 2: Change from "The scale's psychometric properties will be analysed in the following paragraphs, specifically focusing on its reliability and validity" to "The stepwise results of the psychometric evaluation will be proposed in the following". Thanks for the comment. Changed

3.1, para 2: You state "including the minimum value (Min), maximum value (Max), median (Med), mode, and coefficient of variation (Coef Var)" --> I can neither find these values in the table nor are they highlighted. Please adapt the table! Mistake removed

3.1, Tab 1: Change header from "Iten" to "Item" Thanks for the comment. Changed

3.1, Tab1: This might result from the translation process from Portuguese to English, but it seems not to be suitable to formulate QUESTIONS, e.g. "Do you think that learning from this serious game will impact your performance in practice" because this can not properly answered by eg "strongly agree" but rather "yes/no"... Reformulate all items that are questions to STATEMENTS and be sure to provide an initial request to the participants what they should do (e.g. "please assess to which degree you agree with the following statements related to the SEG and mark with an "X""). Otherwise, item formulations and respective anwering options are not consistent and not comprehensible! Thanks for the comment. Changed

3.2: delete (redundant!): "which is extremely high. Con-sequently, the factorability of the correlation matrix is excellent, meaning it is very appro-priate to conduct a factor analysis with these data." Thanks for the comment. Changed

3.2, Tab 2: Items 13, 17, 18 and 19 seem to be double-loading items without showing the suggested difference (approx 0,2) between individual factor loadings to clearly distinguish from each other. You should state in the following text section why these items were  not excluded from the scale and why you chose to allocate them to the respective factors finally. Theoretical reasons? Thanks for the comment. Explain

3.2, Tab 3: Change from "Itens" to "Items" Thanks for the comment. Changed

3.2, Tab 3: change from "Factorial Weight" to "Factor loading" Thanks for the comment. Changed

3.1 AND 3.2 (Tables 1 & 3):  Be consistent in the providion of the formulated items, since both tables show the same items but these items are formulated differently!!! Thanks for the comment. Changed

3.3: Change header from "validity" to "Internal consistency" Thanks for the comment. Changed

3.3: Reformulate from "Subsequently, an analysis of the validity of the questionnaire was conducted, i.e., the internal consistency and reliability of the questionnaire used." to "Subsequently, the internal consistency of the questionnaire was checked by using Cronbach's Alpha coefficients." Thanks for the comment. Changed

3.3, Tab 4: Change from "Dimensao" to "Dimension" Thanks for the comment. Changed

3.3, Tab 4: Change from "Alfa" to "Alpha" Thanks for the comment. Changed

3.3 AND Abstract: Use the coefficients provided in Tab 4 in the Abstract and delete the overall Alpha in the entire Manuscript, since this is not appropriate for the scale. Thanks for the comment. Changed

4 DISCUSSION

4, para 1: this whole section is redundant to the introduction and therefor not suitable for the discussion section; Delete it please! Thanks for the comment. Changed

4, para 1: start with "the aim of this study was to..." and then give a brief overview on the results. Thanks for the comment. Changed

4, para 2,3,4,5,6 are well discussed! Thanks for the comment.

4.1: Please delete this: "Another area for improvement is related to the methodological focus of the study; the study design is pri-marily focused on validating the assessment tool without delving into the effectiveness of serious games as an educational strategy over time." This is not a weakness or limitation! Thanks for the comment. Changed

4.2: Please reformulate or delete: "The development and validation of SEInGAS address the current lack of validated tools for measuring the effectiveness of innovative teaching methods." because the scale can NOT evaluate the effectiveness of SEGs but evaluate its design and assess the presented learning content with relation to learning objectives... Thanks for the comment. Changed

4.2: delete the last sentence (On the other hand, this study also underscores the importance of using validated assessment instruments to evaluate serious games.), because this is not to be seen in this study! Thanks for the comment. Changed

5 CONCLUSION

Please delete the first two sentences or relocate them to the start of the discussion section, where it might be more suitable. Thanks for the comment. Changed

Start with "SEInGAS provides a robust..." Thanks for the comment. Changed

Thank you very much for the relevance of the contributions.

Best regards

Round 2

Reviewer 3 Report

Comments and Suggestions for Authors

Dear authors,

thank you for clearly and dexhaustively addressing the suggested revisions. In common, the manuscript is now well prepared for publication from my point of view. Nevertheless, one issue should be corrected:

RESULTS SECTION:

Table 3 (Scale dimensions): the reformulation "factorial loading" should be applied also to factors 2 & 3

Overall, congratulations, this is a nice paper!

Author Response

Dear Reviewers,

Your insightful comments been invaluable in enhancing the quality and clarity of our work.

Please accept our heartfelt thanks for your contributions, which have undoubtedly made our paper much stronger.

We make the requested changes

Thanks to your detailed feedback.

Best regards
